# Development and Evaluation of a Novel Antibacterial Wound Dressing: A Powder Preparation Based on Cross-Linked Pullulan with Polyhexamethylene Biguanide for Hydrogel-Transition in Advanced Wound Management and Infection Control

**DOI:** 10.3390/polym16101352

**Published:** 2024-05-10

**Authors:** Jiangtao Su, Wantao Yu, Xiaoxia Guo, Chaofan Wang, Qianqiu Wang, Ban Chen, Yuchen Hu, Heshuang Dai

**Affiliations:** 1School of Life and Health Sciences, Hubei University of Technology, No. 28, Nanli Road, Wuhan 430068, China; jiangtsu@hbut.edu.cn (J.S.); wantaoyv1997@163.com (W.Y.); 20200031@hbut.edu.cn (X.G.); wcfhbut@163.com (C.W.); wqqhbut@163.com (Q.W.); chenban@hbut.edu.cn (B.C.); huyuchen@hbut.edu.cn (Y.H.); 2National ‘111’ Center for Cellular Regulation and Molecular Pharmaceutics, Hubei University of Technology, Wuhan 430068, China; 3Key Laboratory of Industrial Microbiology in Hubei, Hubei University of Technology, Wuhan 430068, China; 4Hubei Province Cooperative Innovation Center for Industrial Fermentation, Hubei University of Technology, Wuhan 430068, China

**Keywords:** wound dressing, hydrogel transition, PHMB, pullulan, carboxymethyl cellulose, antibiotic resistance

## Abstract

As antibiotic resistance increasingly undermines traditional infection management strategies, there is a critical demand for innovative wound care solutions that address these emerging challenges. This study introduces a novel antibacterial wound dressing based on Cross-Linked Pullulan (Pul) and Polyhexamethylene Biguanide (PHMB) for enhanced wound management and infection control. The dressing’s adsorption rate reached 200% of its original weight within 30 min, exceeded 300% after 5 h, and exhibited significant non-Newtonian fluid properties. The dressings were able to release the loaded medication completely within 20 min; additionally, the dressing demonstrated significant antibacterial activity against a broad spectrum of bacteria. Significantly, the therapeutic effects of the Pul-PHMB/GP dressing were evaluated in a mouse model. Compared to untreated wounds, wounds treated with Pul-PHMB/GP exhibited a significant gelation process within 5 min post-treatment and showed a significant increase in wound healing rate within 12 days. This powder preparation overcomes the limitations associated with liquid and gel dressings, notably in storage and precise application, preventing the premature expansion or dissolution often caused by PHMB in high-humidity environments. The powder form can transform into a gel upon contact with wound exudate, ensuring accurate coverage of irregular wounds, such as those from burns or pressure sores, and offers excellent chemical and physical stability in a dry state, which facilitates storage and transport. This makes the dressing particularly suitable for emergency medical care and precision therapy, significantly improving the efficiency and adaptability of wound treatment and providing robust support for clinical treatments and emergency responses.

## 1. Introduction

Bacterial infections pose a significant threat to healthcare, necessitating vigilant attention and innovative solutions. Antibiotics have long been the cornerstone of bacterial infection management; however, the emergence of antibiotic resistance has exposed the fragility of this approach [1,2,3]. Thus, it becomes paramount to explore alternative methods for curbing bacterial proliferation [4].

Among various contenders, polyhexamethylene biguanide (PHMB) stands out due to its excellent biocompatibility, antibacterial activity, and low cytotoxicity [5,6,7]. Its superior antibacterial effect is derived from its guanidine structure (Figure 1) [5,8], allowing for highly selective interactions with bacterial cell membranes, disrupting their integrity and leading to cell death [9,10]. Additionally, PHMB can penetrate bacterial cell walls and bind to DNA, causing chromosome damage. Owing to this unique mechanism, PHMB does not easily develop drug resistance. This antibacterial mechanism through electrostatic action makes it difficult for bacteria to develop resistance without altering their entire cell membrane structure.

While PHMB is highly soluble in water and strongly hygroscopic [11], a variety of PHMB-based wound care products, including topical solutions, irrigation fluids, gauzes, and foam-impregnated dressings [6,7,10], are available on the market. When PHMB is directly incorporated into powder-form dressings, it absorbs moisture from the air [5,11], which adversely affects the stability and flowability of the powder, hindering uniform application. Consequently, this limits the practical utility of the dressing for sterile operations and precise application.

According to recent studies, PHMB dressings have shown significant clinical efficacy in reducing infections and promoting wound healing [8,12]. Traditional liquid PHMB dressings, including wound cleansing solution [13], is suitable for initial cleaning and reducing bacterial burden but may require a frequent application to maintain an effective antimicrobial environment, suitable for wounds requiring frequent cleaning and rapid antimicrobial intervention [14]. Gel-form PHMB dressings provide a moist environment for wounds, suitable for keeping wounds moist and applicable to dry or scabbed wounds. However, they are limited by their absorption capacity and observed difficulties in covering wounds [10]. For example, O-chongpian et al. [15] used the TEMPO/NaBr/NaClO system to modify cellulose nanofibers to produce PHMB-loaded hydrogels. Although this formulation allowed for drug release over 3 days, it had limited absorption capacity, with a maximum swelling degree of only about 200% within 24 h. PHMB gauze dressings are commonly used for large-area wounds but require frequent changes if there is significant exudate. These dressings may stick to the wound without enough moisture, potentially causing damage [7]. Laleh et al. [16] developed an alginate–PHMB–AgNPs nanocomposite dressing as a potential alternative. However, it absorbs water quickly and can have inconsistent PHMB loading and release, complicating dosage control and possibly leaving debris in the wound. PHMB fiber dressings [17] are suitable for highly exuding wounds, supporting cell growth and tissue regeneration, and contributing to accelerated wound healing, but require more clinical training and professional knowledge to ensure correct application.

This study introduced an innovative powder preparation wound dressing—Pul-PHMB/GP. The dressing effectively incorporates PHMB from powder form, transforming into hydrogels upon wound application, and exhibits excellent storage stability. The dressing combines sodium alginate for superior water absorption and bio-adhesion [18,19], pectin for added cushioning [19,20], and polyethylene glycol (PEG) to improve mechanical strength and moisture retention [21], enhancing adaptability to meet varied clinical needs and promote wound healing effectively. This powder preparation excels in wound healing by maintaining a moist environment, reducing inflammation, and promoting tissue regeneration. It overcomes the limitations associated with liquid and gel dressings, notably in storage and precise application, preventing the premature expansion or dissolution often caused by PHMB in high-humidity environments. The powder form can transform into a gel upon contact with wound exudate, providing lasting antimicrobial protection while reducing the frequency of changes; meanwhile, it ensures accurate coverage of irregular wounds, such as those from burns or pressure sores, and offers excellent chemical and physical stability in a dry state, which facilitates storage and transport. When applied directly to wounds without the need for additional solvents, the dressing provides immediate antimicrobial protection. The gels formed not only retain their shape but also absorb significant amounts of water or biological fluids, creating a protective viscous layer that enhances the healing environment [22].

During the investigation of adsorption experiments with simulated wound fluid, it was revealed that Pul-PHMB/GP absorbed up to 200% of its original weight within 30 min and exceeded 300% after 5 h, demonstrating exceptional adsorptive performance suitable for wound dressing applications. Rheologically, both Pul-PHMB/GP and CMC-PHMB/GP dressings displayed significant non-Newtonian fluid characteristics, with viscosity decreasing as shear rates increased. This feature ensures that the dressings maintain a gel-like state under minimal external forces, such as gravity or clothing friction, making them ideal for forming a protective layer over diverse wound surfaces. The bacteriostatic effectiveness of the PHMB-loaded powder dressing was also validated, showing substantial antibacterial activity against a wide range of bacteria, including *Pseudomonas aeruginosa*, *Bacillus subtilis*, *Staphylococcus aureus*, and *Escherichia coli*. Additionally, the therapeutic effects of the Pul-PHMB/GP dressing were assessed in a mouse model. Compared to untreated wounds, wounds treated with Pul-PHMB/GP underwent a significant gelation process within 5 min of application and demonstrated a marked improvement in healing rates within 12 days. This rapid gelation and enhanced healing rate highlight the dressing’s effectiveness in managing and expediting wound recovery. This makes the dressing particularly suitable for emergency medical care and precision therapy, significantly improving the efficiency and adaptability of wound treatment and providing robust support for clinical treatments and emergency responses.

## 2. Material and Methods

### 2.1. Materials

#### 2.1.1. Equipment

A UV/VIS Spectrophotometer (ZF1-П, Shanghai, China) and a Vacuum Freeze Dryer (FD-2C-80, Shanghai, China) were used in this study. Additionally, a Transdermal Diffusion Meter (TPY-2, Shanghai, China), a Laser Particle Size Analyzer (LT2200, Zhuhai, China), an FTIR Spectrometer (Thermo Fisher IS10, Waltham, MA, USA), an SEM Microscope (JSM6390LV, Tokyo, Japan), and an Optical Microscope (DXS-2, Shanghai, China) were employed.

#### 2.1.2. Chemicals and Reagents

Polyhexamethyl Biguidine (PHMB, 98%) was sourced from Changsha Yanbang Chemical Technology Co. LTD, Changsha, China. Sodium Alginate (99%), Polyethylene Glycol 8000 (PEG8000, 99%), Sodium Carboxymethyl Cellulose (CMC, 99%), and Pullulan (99%) were acquired from Huate Sheng Biotechnology (Wuhan) Co., Ltd. in Hubei, China. N, N-Methylene Bisacrylamide (MBA, 99%) and Methacrylic Acid (MA, 99%) were procured from China National Pharmaceutical Group Corporation Chemical Reagent Co., Ltd., Beijing, China.

#### 2.1.3. Animal Model

Mice (Kunming SPF, weight 18–22 g) were obtained from the Hubei Experimental Animal Research Center.

### 2.2. Preparation of CMC-Gel and Pul-Gel Particles

Dissolve 0.75 g of CMC in 35 mL of purified water. Stir this mixture in a three-necked flask maintained at 70 °C until complete dissolution. Prepare a potassium persulfate solution by dissolving 0.3 g (0.0011 mol) in 10 mL of pure water. Add the potassium persulfate solution dropwise to the CMC solution, stirring continuously at 70 °C for 10 min. Cool the flask to room temperature. Prepare a mixture solution by dissolving 0.0537 g of MBA (0.0003 mol) into 30 g of MA (0.35 mol). Add this mixture to the flask while stirring continuously. Add 24 mL of purified water to the flask and heat it to 60 °C in an oil bath. Once the solution becomes viscous, transfer it to a beaker and maintain the reaction at 60 °C for 2 h. Cool the resultant mixture to room temperature to obtain hydrogel, which is highly absorbent. Cut the gel into small pieces and soak them in water to allow for swelling. Immerse the hydrogel in a 0.1 M NaOH solution for purification. Continued soaking until the pH reaches a neutral level. Freeze-dry the gel at −50 °C and pulverize it using a ball mill to obtain CMC–gel particles.

Replace CMC with Pul and repeat the steps above to obtain the new particle (Pul–gel).

### 2.3. Response Surface Methodology

The formulation of in situ hydrogels was optimized through the Central Composite Design (CCD) method by Design-Expert 13. In the experiment, the proportion of sodium alginate was set at 10%, 20%, and 30%, and the proportion of pectin was set at 5%, 10%, and 15%, serving as the two main independent variables for response surface optimization. The performance of different formulations was assessed by measuring fluid uptake, viscosity, and Hausner ratio. The optimal formulation was determined through response surface analysis. The experimental design is shown in Table 1.

### 2.4. Preparation of Composite Dressing Particles

Blended particles were prepared by ball milling sodium alginate (2 g), pectin (1 g), PEG8000 (6 g), and PHMB-loaded gel particles (0.47 g) for 30 min at 300 rpm/min. The dried composite dressing particles were obtained after passing through a 200-mesh sieve. Figure 2 visually illustrates the key stages of the sequential process involved in synthesizing our innovative hydrogel-based wound dressings. Commencing with the foundational polysaccharide, the process encompasses cross-linking, water absorption, blending, crushing, and two stages of freeze-drying. These sequential steps play a pivotal role in attaining the distinctive attributes of the powder preparation dressings by Pul–gel and CMC–gel and are specifically engineered to transform from powder to hydrogel upon application to augment their efficacy in wound management.

### 2.5. Swelling Experiments

We prepared 1000 mL of 0.1 M HCl solution (pH = 1.2) and 1000 mL of phosphate buffer saline (PBS, 0.1 M, pH = 7.4). Six pieces of dried hydrogel obtained from Section 2.2 were prepared, each weighing approximately 0.55 g. Three pieces were immersed in 100 mL HCl solution, and the other three in 100 mL PBS. The weight change was measured at fixed intervals until it became constant, indicating that the hydrogels had reached swelling equilibrium. The average weight of the three sets of data was used to plot the swelling curve. The suitability and performance of these hydrogels were compared with commercial wound dressings using the most common parameters for quantifying swelling capacity.

### 2.6. Preparation of PHMB-Absorbed Gel Particles

We dissolved 100 mg of PHMB in 10 mL of pure water to create an aqueous drug solution. The freeze-dried hydrogel particles (0.5 g) were soaked in this PHMB solution. Once the gel absorbed the solution completely in a sealed beaker, they were freeze-dried again at −80 °C. After 30 min of ball milling at 300 rpm/min, the PHMB-absorbed gel particles were obtained through a 200-mesh sieve.

### 2.7. FT-IR Measurements

FT-IR spectra were recorded using a Bruker Tensor spectrometer. KBr pellet samples were prepared that included both gel particles and PHMB-loaded particles.

### 2.8. Morphology Observation of PHMB-Loaded Gel Particles

The surface of the dried drug-loaded gel was coated with gold, and its microstructure was observed using a scanning electron microscope (SEM).

### 2.9. Determination of Drug Loading Efficiency

Dried PHMB-loaded gel particles (5 mg, W) were dispersed in a phosphate buffer solution (5 mL, pH = 7.4) and then placed into a dialysis tube. The sealed dialysis tube was immersed in a conical flask containing phosphate-buffered saline (PBS, 30 mL, pH = 7.4) and maintained at 37 °C for 6 h. Subsequently, the concentration (C) and weight (W1) of released PHMB were determined using a UV-vis Spectrophotometer at 236 nm, and these values were calculated using the standard curve method. Finally, the drug loading efficiency (DLE) was calculated according to Equation (1).
(1)$$\mathrm{DLE}\%=(\frac{W1}{W2})\times 100\%$$

### 2.10. Kinetic Study of PECF Absorption in Blending Powder Dressing Particles

A pseudo-extracellular fluid (PECF) was utilized to investigate the water absorption kinetics of the blending dressing particles. The preparation of the PECF solution was carried out in accordance with the methods detailed in report [23]. The receptor chamber of the Vertical Diffusion Cell (VDC) was filled with the PECF solution. A 40 mg sample of blending dressing particles was placed on the cellulose membrane in the donor chamber at a constant temperature of 37 °C. At designated time intervals, the total weight of the donor chamber was measured, and PECF was replenished in the receptor chamber to maintain a constant total volume of 8 mL.

### 2.11. Rheological Properties of Blending Powder Dressing Particles

Blending dressing particles (1 g) were mixed with 3.5 mL of PECF solution. Ultrasound treatment yielded a hydrogel. This hydrogel was allowed to stand for 15 min to reach thermal equilibrium. Rheological studies were conducted at shear rates ranging from 1 S^−1^ to 1000 S^−1^. The angular frequency was fixed at 10 rad/s, and strain amplitudes varied from 0.0625% to 500%.

### 2.12. Physical Characteristics of Blending Powder Dressing Particles

The Hausner ratio was calculated based on the following formula: Hausner Ratio = V2/V1. The blending dressing particles (40 mg) were tapped gently in a 1 mL syringe to record the initial volume (V1) and continued tapping until no further change in volume (V2) was observed, representing the loose and tapped densities, respectively. Laser particle size analysis and microscopic evaluation were carried out as independent sub-sections to provide additional characterization data.

### 2.13. In Vitro Release Studies of PHMB

Blending dressing particles (10 mg) were put into dialysis bags in triplicate, respectively, by adding PBS (1 mL, pH = 7.4) to disperse. The dialysis bag was placed in a conical flask containing PBS (30 mL, pH = 7.4) to be released at a constant 37.0 °C. The dialysate was sampled at 5 min, 10 min, 15 min, 20 min, 30 min, and 40 min, respectively. The release percentage of PHMB was calculated by the concentration of PHMB in the dialysate.

### 2.14. Determination of Antibacterial Activity

*Staphylococcus aureus*, *Pseudomonas aeruginosa*, *Bacillus subtilis*, and *Escherichia coli* were selected as experimental strains and bacterial suspensions were prepared with concentrations from 1 × 10^8^ CFU/mL to 5 × 10^8^ CFU/mL. Blending powder dressing and the bacteria suspension were added to the medium. The living bacteria started to count after being cultured for 24 h. The samples were cultured in an incubator at 37.0 °C for 48 h. The experiment was repeated 3 times, and the concentration of living bacteria (CFU/mL) in each group was converted to log values.

### 2.15. In Vivo Experiments on Mice

*Pseudomonas aeruginosa* induced to the cortex injury infection model in mice was established. PHMB-loaded powder dressing (Pul-PHMB/GP) was selected as a candidate according to the comparison results to evaluate the healing ability of infected wounds. Simulation of the wound was prepared and inoculated with *pseudomonas aeruginosa* suspension at a concentration of 10^8^ CFU/mL with a diameter of about 10 mm. After 2 days, the infected mice were treated with PHMB-loaded powder dressing (Pul-PHMB/GP) and non-drug-loaded powder dressing (Pul/GP). The blank controller was set by untreated infected mice. We took photos every two days.

### 2.16. Histopathology

Skin samples were excised from the dorsal side of mice using a scalpel and immediately fixed in 10% formaldehyde. After 24 h of fixation, the samples were rinsed twice with PBS solution (pH 7.4), with each rinse lasting 5 min. Subsequently, the tissues were processed into paraffin and sectioned at a thickness of 6 μm using a microtome. The paraffin-embedded tissue sections were then dewaxed and rehydrated, followed by staining with Hematoxylin and Eosin (H&E).

### 2.17. Ethical Considerations in Animal Experimentation

In accordance with ethical standards for animal experimentation, all procedures involving mice in this study were rigorously reviewed and approved by the Institutional Review Board of Hubei University of Technology. The approval for the experimental protocol, identified under the code HBUT20230085, was granted on 9 June 2023. This approval ensured that all experimental practices, including Section 2.15 and Section 2.16 of the study.

## 3. Results and Discussion

### 3.1. The PHMB Loading and Release Mechanism of the Dressing

In this study, we synthesized two types of novel antibacterial dressings through specific steps. As shown in the reaction equation in Figure 3, each type of dressing was treated with polysaccharides in the presence of a free radical initiator, potassium persulfate, along with N,N′-methylenebisacrylamide (MBA), and methacrylic acid (MA), reacting at 60 °C for 2 h to form a cross-linked gel network.

Subsequently, polyhexamethylene biguanide (PHMB) was introduced into the cross-linked network. Due to the presence of amino and guanidine groups in PHMB, hydrogen bonds can form with the hydroxyl groups on the polymer chains, aiding in stabilizing the distribution of PHMB within the dressing. The multiple guanidine groups (-NH-C(=NH)-NH-) in PHMB carry a positive charge, enabling electrostatic attraction with negatively charged carboxyl groups (-COO-). In the case of CMC, its carboxymethyl side chains provide electrostatic interaction points with the positively charged guanidine groups of PHMB. This structure not only allows for uniform distribution of PHMB but also its gradual release, maintaining prolonged antibacterial activity as the dressing slowly degrades, thereby ensuring sustained antimicrobial effectiveness.

### 3.2. Analysis of Response Surface Optimization Results

Figure 4 presents various three-dimensional response surface plots observed by adjusting the ratios of sodium alginate and pectin. This study utilized the Central Composite Design (CCD) method to optimize the formulation of in situ hydrogels, investigating the impact of sodium alginate and pectin concentrations on hydrogel performance through Response Surface Methodology (RSM). The response surface for fluid uptake (Figure 4A) indicated a significant synergistic effect of the two materials, particularly at higher concentrations. The highest fluid uptake was noted when the sodium alginate concentration was approximately 25−30%, and the pectin concentration was about 10−15%. The Hausner ratio, serving as an indicator of powder flowability (Figure 4B), was presented on a response surface that exhibits a relatively flat trend but inclines towards one side. This inclination indicated a notable rise in the Hausner ratio with higher concentrations of sodium alginate and pectin, suggesting reduced flowability at elevated concentrations. Furthermore, the viscosity response surface (Figure 4C) demonstrated a distinct upward trend, indicating that increases in the concentrations of sodium alginate and pectin positively affect the viscosity. The mixture’s viscosity was notably enhanced, especially at higher concentrations. Figure 4D shows the results of the desirability analysis after overall performance optimization. The desirability index peaks at mid-range concentrations, the optimal concentrations of sodium alginate and pectin, approximately 22.1% and 11.7%, respectively, which are within their optimal operating ranges (sodium alginate 10−30%, pectin 5−15%).

### 3.3. FT-IR Spectra Analysis

Figure 5 illustrates the FT-IR spectra of Pul–gel and CMC–gel. A ν_-OH_ bond was detected at 3700 cm^−1^. Stretching vibrations relative to the -OH groups were detected at 3700 cm^−1^. Additionally, a ν_C-H_ bond appeared at approximately 2933 cm^−1^, and C-O bonds were observed at about 1089 cm^−1^. Strikingly, the disruption in the ν_-OH_ (3700 cm^−1^) bond post-PHMB addition suggests that the intrinsic hydrogen bonding network within the gel particles was altered. This effect could be a result of newly established hydrogen bonds or van der Waals interactions between PHMB and the carboxyl groups, which potentially decreased the interactions of -OH with other functional groups in the gel matrix.

### 3.4. Swelling Properties of Pul–Gel and CMC–Gel

Upon investigation of the pH-responsive swelling behaviors of Pul–gel and CMC–gel, a distinct discrepancy emerged (Figure 6). Both hydrogels exhibited diminished swelling ratios under acidic environments compared to neutral ones. Strikingly, Pul–gel consistently lagged behind CMC–gel in terms of swelling ratio when tested under identical pH conditions.

This pronounced disparity can be largely attributed to two key factors. First, CMC–gel boasts a higher density of carboxyl functional groups compared to Pul–gel. These carboxyl groups tend to ionize under neutral and basic conditions, generating a greater net negative charge that amplifies electrostatic repulsion between the gel’s polymeric chains. Second, the higher density of polar or ionizable groups in CMC–gel creates a more favorable environment for water molecules to surround and interact with the gel, facilitating a more efficient solvation process. These collective attributes act in concert to amplify the gel’s capacity for water absorption and retention, thereby enhancing its overall swelling ratio.

### 3.5. Morphological Study of Pul–Gel and CMC–Gel

After undergoing freeze-drying, the skeletal structure of the gels was preserved. Both types of gels exhibited a cavity structure with a regular grid pattern, facilitating the loading of water-soluble drugs (Figure 7). When these gels were utilized for adsorbing PHMB, the substance remained within the cavities even after freeze-drying, crushing, and subsequent blending. The retention of PHMB within the cavities may be attributed to its interactions with specific functional groups in the gel, such as carboxyl and hydroxyl groups, thereby achieving efficient adsorption and release. These findings not only offer a new avenue for the controlled release of water-soluble drugs but also provide crucial experimental evidence for further research into the potential biomedical applications of gel materials.

### 3.6. Analysis of Pul-PHMB/GP and CMC-PHMB/GP

The detailed procedures for preparing Pul-PHMB/GP and CMC-PHMB/GP powder dressings were outlined in Section 2.6. Microscopic observation revealed that the particles were homogeneous (Figure 8). Table 2 demonstrates that the blending, crushing, and sieving process allowed for the even mixing of the gel particles with other auxiliary materials. The Hausner ratio for Pul-PHMB/GP powder dressing was approximately 1.68, while the ratio for CMC-PHMB/GP powder dressing was markedly higher at 2.71, attributed mainly to its hygroscopic properties.

This significant difference in Hausner ratios implies that the fluidity of CMC-PHMB/GP powder dressing is compromised, which is problematic for its effective application on wounds. In contrast, Pul-PHMB/GP powder dressing retains better fluidity, deeming it more suitable for wound applications. The choice of gel substrate can influence the Hausner ratio due to differences in particle shape and size distribution, and gel-based pullulan may offer better fluidity owing to weaker inter-particle interactions. Moreover, the higher density of functional groups in CMC could increase particle interactions, thereby reducing fluidity. Incorporating all these findings, it is concluded that gel-based pullulan is the more advantageous choice for the preparation of powder dressings with satisfactory fluidity.

### 3.7. Adsorptive Property of Blended Powder Dressing Particles

The adsorptive capability of the powder dressing toward simulated wound fluid demonstrated remarkable efficacy in the experiment (Figure 9). Specifically, the adsorption rate reached 200% of its original weight within a mere 30 min and exceeded 300% in 5 h. This impressive adsorptive performance essentially meets the application requirements for wound dressings. These may be due to the polymers used in the experiment—carboxymethyl cellulose (CMC) and polyacrylic acid (Pul)—are replete with functional groups that can form stable hydrogen bonds with water molecules, thus contributing to their high absorptivity. Furthermore, the addition of the antibacterial agent PHMB, with its cationic characteristics, generated electrostatic interactions with the negatively charged groups in CMC and Pul, thereby further enhancing the material’s adsorptive capacity.

### 3.8. Rheological Properties of Blended Powder Dressing Particles

In this study, powder dressings were found to form adhesive gels upon absorption of wound exudate, thereby maintaining a moist environment conducive to wound healing. Rheological tests revealed notable differences between the two types of gels (CMC and PUL) in terms of viscosity, shear stress, storage modulus (G′), and loss modulus (G″).

The steady-state shear viscosity of both gels sharply decreases with increasing shear rate, highlighting their non-Newtonian flow characteristics (Figure 10A). This suggests that both gels are capable of maintaining a gel-like state at the wound site, resisting weak shear stresses such as gravity or clothing friction (Figure 10B). This may be due to the high density of functional groups in CMC leads to enhanced inter-particle interactions, resulting in higher storage modulus (G′) and more elastic (solid-like) behavior under low strain amplitudes (Figure 10C,D). Conversely, Pul displays greater viscous (liquid-like) behavior due to weaker inter-particle interactions.

Further increases in strain amplitude result in a more significant drop in storage modulus (G′) over loss modulus (G″), especially evident in Pul gels. This could be because Pul molecular chains are more flexible, easily undergoing conformational changes under high strains, thereby reducing the gel’s adhesive properties. Incorporating these findings, we conclude that Pul-based gels are more suited for the creation of powder dressings with satisfactory fluidity.

### 3.9. Loading and Release Behavior of CMC-PHMB/GP and Pul-PHMB/GP

The amount of PHMB released was quantified using a UV spectrophotometer at a wavelength of 236 nm. Absorbance (Y) was proportional to the concentration of PHMB (x) and was calculated using the Equation (2).
Y = 0.05125x + 0.00284 (R^2^ = 0.99915)(2)

The drug loading efficiency (DLE) for CMC–gel and Pul–gel was 23.4 ± 0.13% and 20.7 ± 0.38%, respectively. These variances could be attributed to their distinct chemical structures and crosslinking densities. The higher DLE observed in CMC–gel is likely due to its greater functional groups and crosslinking density compared to Pul–gel, which has a relatively looser structure. The DLE for CMC-PHMB/GP and Pul-PHMB/GP were 1.2 ± 0.011% and 1.02 ± 0.06%, respectively. Figure 11 indicated that PHMB was fully released from both types of dressing particles within 20 min.

### 3.10. Bacteriostatic Effect of PHMB-Loaded Powder Dressing

Figure 12 illustrates the killing log-values of the PHMB-loaded powder dressing against Gram-positive bacteria, including *Pseudomonas aeruginosa*, *Bacillus subtilis*, and *Staphylococcus aureus*, as well as *Escherichia coli*, which belongs to Gram-negative bacteria, were all greater than 4. The results demonstrated a significant difference compared to the blank group (blended hydrogel samples without PHMB loading) with *p* < 0.05, revealing a remarkable antibacterial effect against a broad spectrum of bacteria by the fresh dressing.

### 3.11. Evaluation in Model Mice

Because the CMC-PHMB/GP formulation exhibited substantial hygroscopic properties, it led to a sticky and unmanageable consistency after storage. This condition rendered the CMC-PHMB/GP unsuitable for further processing and clinical testing. Consequently, our focus shifted exclusively to Pul-PHMB/GP, which maintained desirable physical properties conducive to both experimental handling and potential therapeutic application. In the model mice evaluation, Pul-PHMB/GP demonstrated significant effectiveness in promoting wound healing and controlling infections. Compared to untreated infected wounds, a significant gelation process was observed within 5 min of applying the powder dressing (Figure 13).

Notably, the Pul-PHMB/GP dressing significantly expedited the wound recovery process (Figure 14 and Figure 15). Statistical tests revealed a significant difference in the wound healing rate (*p* < 0.05).

In this study, we evaluated the impact of Pul-PHMB/GP on wound healing by comparing normal skin tissue, untreated wound tissue, and wound tissue treated with Pul-PHMB/GP. Figure 16A shows the tissue morphology of normal skin, which is clear and wellintegrated. Figure 16B shows the untreated wound tissue, which clearly exhibits epidermal loss and, compared to normal skin, has not effectively recovered, remaining in a damaged state. In contrast, Figure 16C shows a tissue section of a wound treated with Pul-PHMB/GP, which resembles the normal skin structure, indicating effective recovery of the skin tissue after treatment. These results demonstrate the significant positive effects of Pul-PHMB/GP on wound treatment, effectively facilitating the recovery of skin tissue to near-normal conditions, while untreated wounds continue to show persistent tissue damage.

## 4. Conclusions

This study offers a comprehensive investigation into wound dressings based on CMC and Pul, displaying pronounced advantages in absorbency, rheological properties, PHMB loading and release behavior, and antibacterial activity. Intriguingly, the choice of CMC and Pul as base materials manifest different characteristics across various performance metrics, which can be attributed to their inherent chemical structures and molecular compositions.

CMC’s chemical architecture lends it greater absorbency, possibly due to an increased number of hydroxyl groups that provide additional adsorption sites. On the other hand, Pul exhibits more robust rheological properties, likely owing to the flexibility and elasticity of its molecular chains.

In terms of PHMB loading and release, both materials displayed differences that were potentially related to their respective pore structures and surface chemistries. These variances broaden the range of potential applications and customizations for the dressings. Both materials showed significant antibacterial effects in antimicrobial assays. However, these effects varied across different types of bacteria, further affirming the multifunctional and broad-spectrum nature of the dressings.

In summary, this study offers an in-depth and holistic understanding of why CMC and Pul provide a scientific basis for further design and optimization of dressings. In order to enhance the capabilities of the Pul-PHMB/GP wound dressing, future research would focus on optimizing the release kinetics of PHMB to extend its antibacterial effects and on integrating various therapeutic agents to create multifunctional wound care solutions. These advancements would be aimed at expanding the clinical utility of the dressing by potentially incorporating bioactive compounds that accelerate tissue regeneration and alleviate pain. This forward-looking development will utilize the latest advancements in materials science and biotechnology to address the evolving demands of modern wound care.

## Figures and Tables

**Figure 1 polymers-16-01352-f001:**
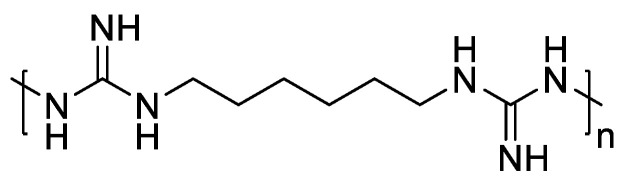
Chemical structure of PHMB.

**Figure 2 polymers-16-01352-f002:**
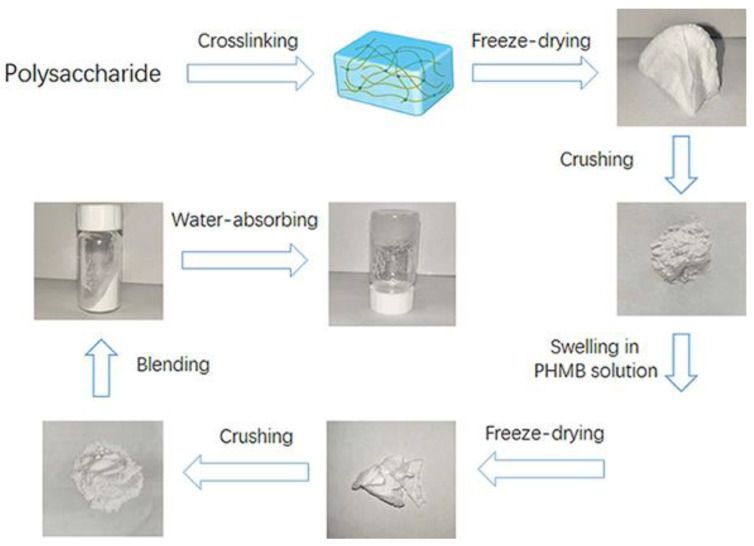
Preparation process of two types of wound dressing.

**Figure 3 polymers-16-01352-f003:**
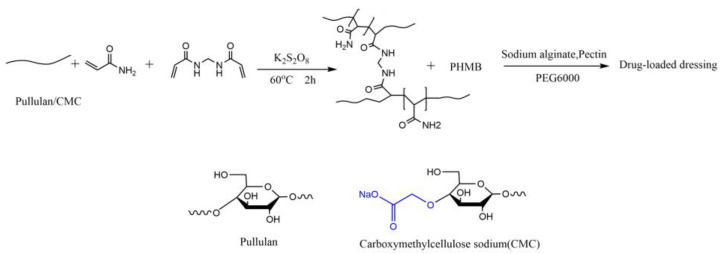
Synthesis scheme of PHMB-loaded wound dressing.

**Figure 4 polymers-16-01352-f004:**
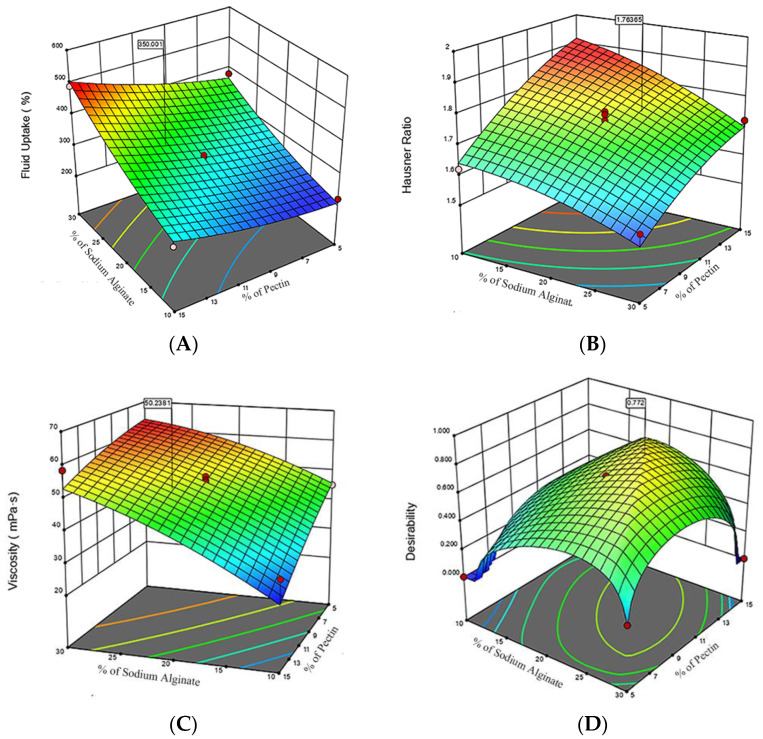
Composite Response Surface Plots for Hydrogel Formulation Optimization (**A**) Fluid Uptake Response Surface: demonstrates the impact of sodium alginate and pectin concentrations on the hydrogel’s fluid uptake percentage; (**B**) Hausner Ratio Response Surface: demonstrates the impact of sodium alginate and pectin concentrations on the hydrogel’s Hausner ratio; (**C**) Viscosity Response Surface: illustrates the relationship between polysaccharide concentrations and viscosity; (**D**) Desirability Response Surface: depicts the optimal desirability scores achieved with moderate concentrations of sodium alginate and pectin. The red dots represent the measured data points under specific experimental conditions.The different colored regions in the figure represent areas of different optimality levels, with regions of higher optimality typically shown in darker colors, while regions of lower optimality are shown in lighter colors.

**Figure 5 polymers-16-01352-f005:**
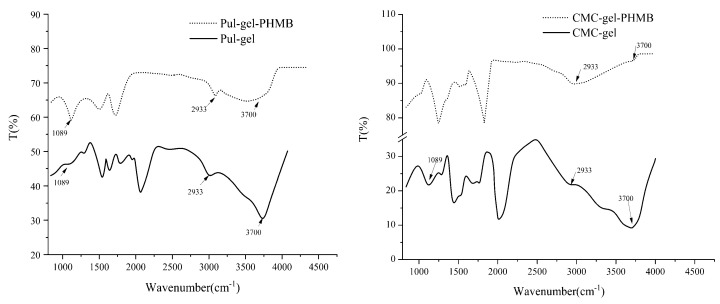
FT-IR spectra of gel particles and PHMB-loaded powder dressing.

**Figure 6 polymers-16-01352-f006:**
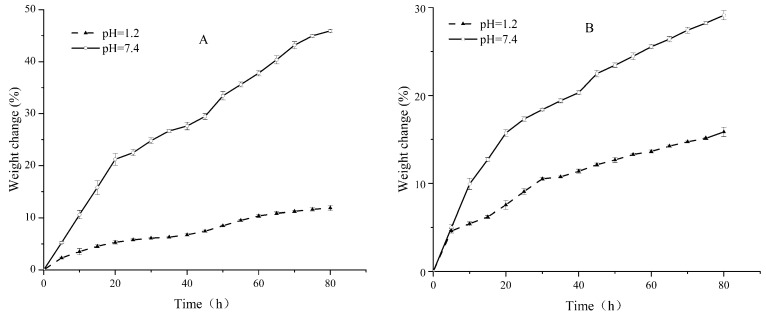
Swelling equilibrium curves at different pH values (**A**) Pul–gel; (**B**) CMC–gel.

**Figure 7 polymers-16-01352-f007:**
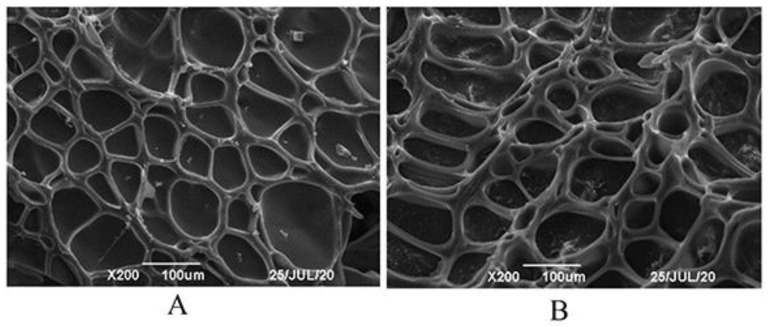
SEM images of dried gel particles (**A**) Pul–gel; (**B**) CMC–gel.

**Figure 8 polymers-16-01352-f008:**
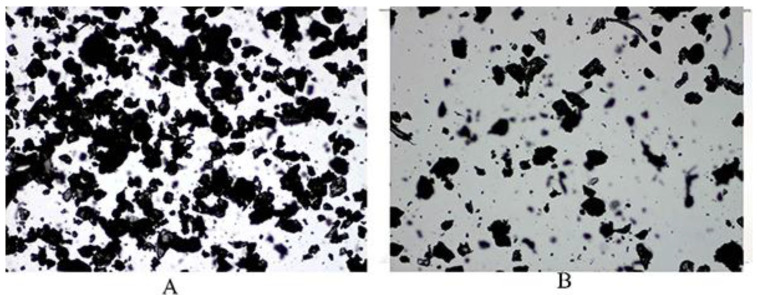
The morphology of blending PHMB-loaded dressing at 4 times magnification under the microscope: (**A**) Pul-PHMB/GP; (**B**) CMC-PHMB/GP.

**Figure 9 polymers-16-01352-f009:**
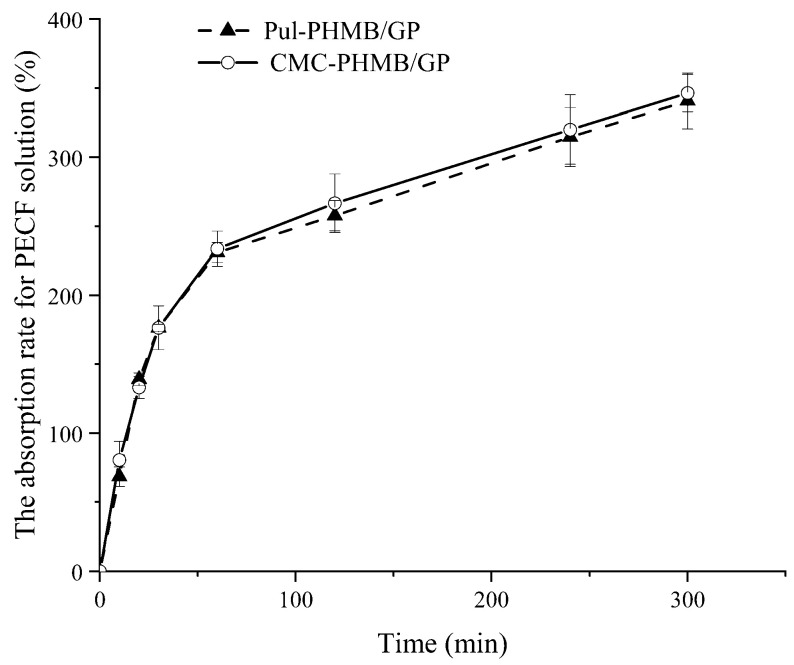
The absorption rate of Pul-PHMB/GP and CMC-PHMB/GP for PECF solution within 5 h.

**Figure 10 polymers-16-01352-f010:**
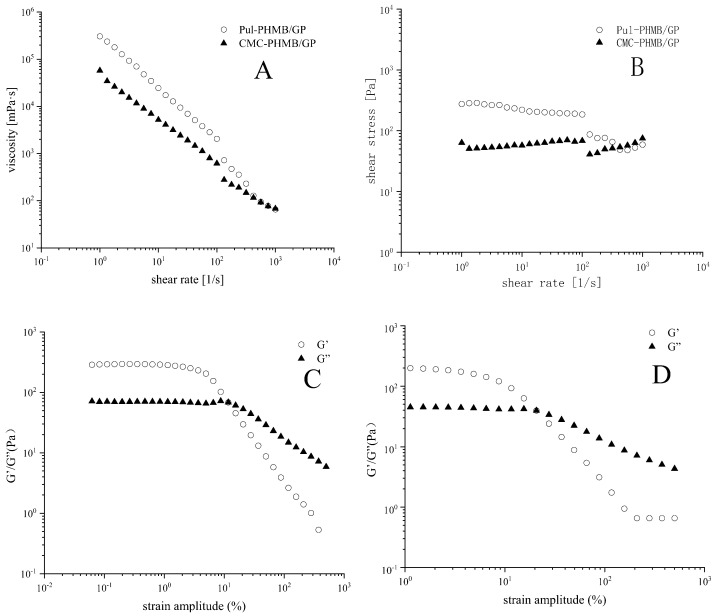
(**A**) The relationship between viscosity and the shear rate of Pul-PHMB/GB and CMC-PHMB/GB; (**B**) relationship between shear stress and the shear rate of Pul-PHMB/GB and CMC-PHMB/GB; (**C**) relationship between strain amplitude and the storage modulus G′ and loss modulus G″ of Pul-PHMB/GB; (**D**) relationship between strain amplitude and the storage modulus G′ loss modulus G″ of CMC-PHMB/GB.

**Figure 11 polymers-16-01352-f011:**
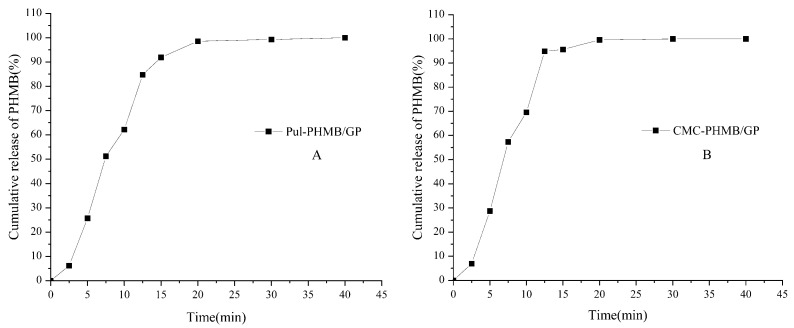
Cumulative release of PHMB (%) in PBS (pH = 7.4) by powder dressing particles: (**A**) Pul-PHMB/GP; (**B**) CMC-PHMB/GP.

**Figure 12 polymers-16-01352-f012:**
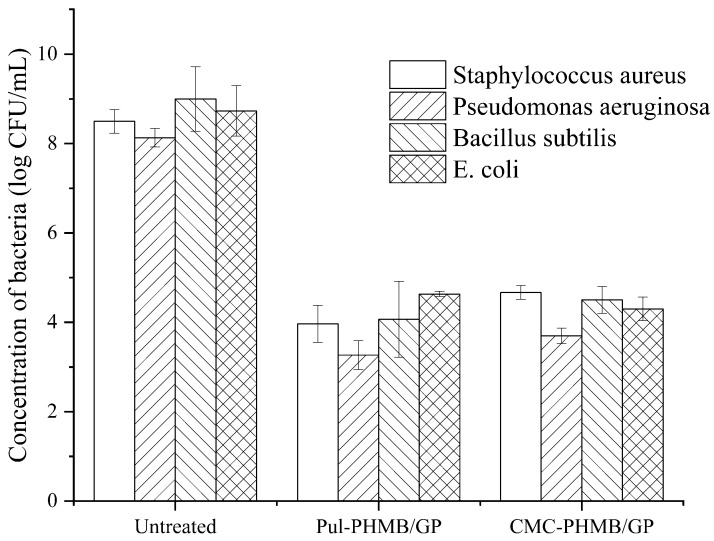
Antimicrobial effect of Pul-PHMB/GP and CMC-PHMB/GP dressings on *Pseudomonas aeruginosa*, *Bacillus subtilis*, *Staphylococcus aureus*, and *Escherichia coli*.

**Figure 13 polymers-16-01352-f013:**
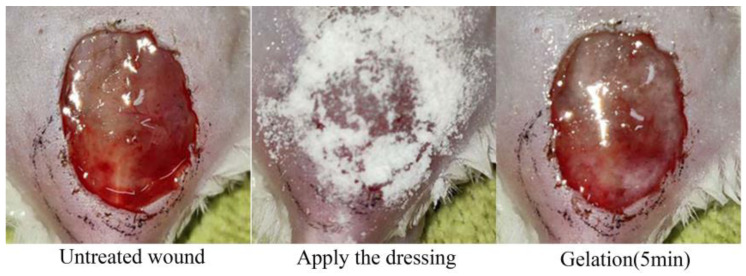
Gelation of dressing on the wound surface.

**Figure 14 polymers-16-01352-f014:**
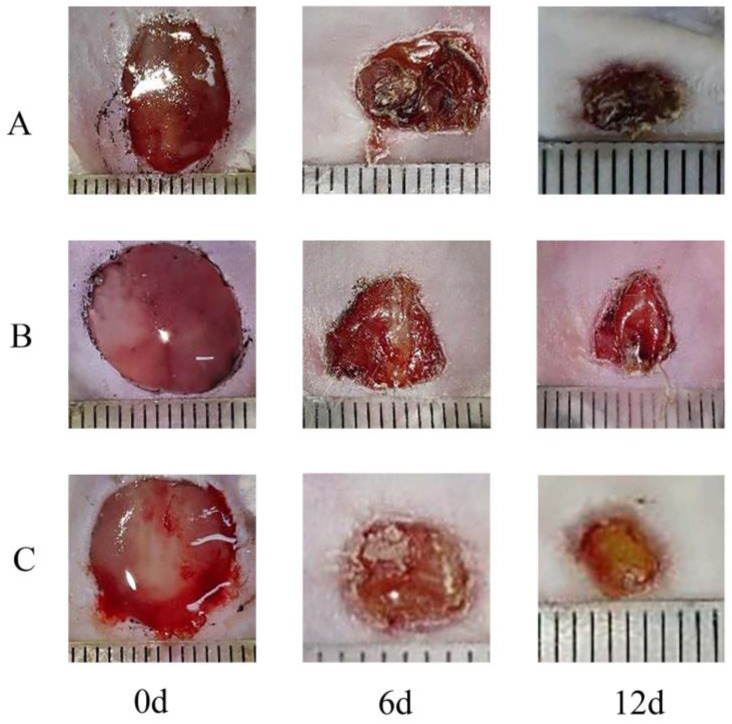
Comparison of wound area between Pul-PHMB/GP dressing and non-loading dressing at 0 days, 6 days, and 12 days. (**A**) untreated wound; (**B**) wound treated by Pul/GP; (**C**) wound treated by Pul-PHMB/GP.

**Figure 15 polymers-16-01352-f015:**
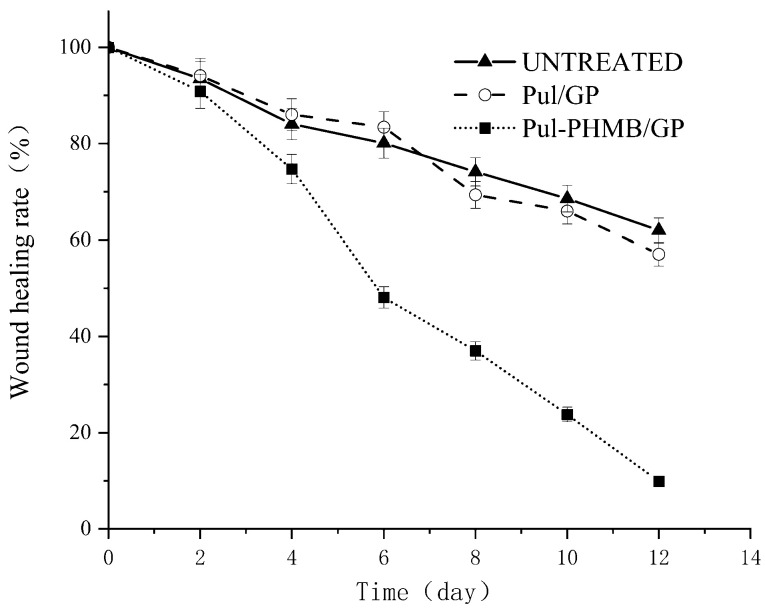
Graph of the healing effect of PHMB-loaded dressing (Pul-PHMB/GP) and non-loaded dressing (Pul/GP) and the untreated.

**Figure 16 polymers-16-01352-f016:**
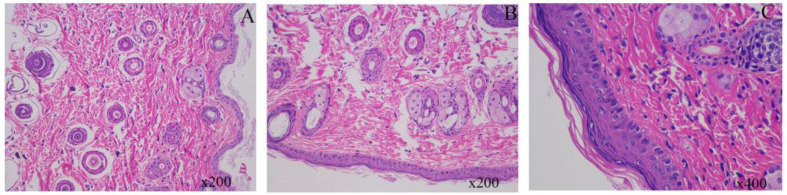
Histological Analysis of Mouse Skin Tissue Under Various Treatment Conditions. (**A**) normal mouse skin tissue; (**B**) untreated mouse skin tissue after wound infection for 12 days; (**C**) mouse skin after wound infection treated with Pul-PHMB/GP for 12 days.

**Table 1 polymers-16-01352-t001:** Experimental Design and Results for Hydrogel Optimization.

Trial No.	Content (%)			Result		
	Sodium Alginate	Pectin	PEG6000	Fluid Uptake (%)	Viscosity (mPa·S)	Hausner Ratio
1	0.2	0.1	0.7	350	58.2	1.75
2	0.3	0.05	0.65	380	61.4	1.55
3	0.3	0.15	0.55	550	66.1	1.72
4	0.2	0.1	0.7	351	54.1	1.78
5	0.05	0.1	0.8	238	24.6	1.86
6	0.2	0.1	0.7	344	52.9	1.77
7	0.34	0.1	0.5	526	66.4	1.53
8	0.2	0.1	0.6	481	28.1	1.86
9	0.1	0.05	0.85	242	45.1	1.62
10	0.2	0.02	0.7	284	62.8	1.51
11	0.1	0.15	0.75	300	31.4	1.88
12	0.2	0.1	0.7	361	60.2	1.74
13	0.2	0.1	0.7	340	64.2	1.71

**Table 2 polymers-16-01352-t002:** Data were obtained by Laser Particle Sizer and calculated.

Sample	Mean Size/μm	Angle of Repose/°	Hausner Ratio
Pul-PHMB/GP	58.66	34.44	1.68
CMCPHMB/GP	54.02	38.11	2.71

## Data Availability

The original contributions presented in the study are included in the article, further inquiries can be directed to the corresponding author.

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
