# Peer review of "Development and Evaluation of a Novel Antibacterial Wound Dressing: A Powder Preparation Based on Cross-Linked Pullulan with Polyhexamethylene Biguanide for Hydrogel-Transition in Advanced Wound Management and Infection Control"

_polymers, 2024, doi:10.3390/polym16101352_

Round 1

Reviewer 1 Report

Comments and Suggestions for Authors

Review

Manuscript ID:  polymers-2957153

Journal: polymers

 Development and Evaluation of a Novel Pul-PHMB/GP Anti- 2 bacterial Wound Dressing: A Cross-Linked, Hydrogel-Transition Solution for Advanced Wound Management and Infection  Control

The authors try to study a novel antibacterial wound dressing, Pul-PHMB/GP, synthesized  by cross-linking pullulan with sodium carboxymethyl cellulose and incorporating Polyhexamethylene Biguanide (PHMB). This development comes as an innovative response to the growing concern over antibiotic resistance, featuring a unique mechanism that enables it to transition from powder to hydrogel upon application, thus offering both optimal storage conditions and adaptability to wound environments.. But there are some comments as the following:

1-    Title must be modified to remove abbreviation

2-    In abstract, the authors should add problem at first of abstract

3-    In abstract, authors should add more quantitative data

4-    In abstract, the authors should add more data at least 250-300 words

5-    In abstract, the authors must add prospective work at end of abstract

6-    In abstract, the authors must add novelty in abstract

7-    Keywords must at least six words

8-    Page 2 line 62,authors must move this into experimental

9-    Figures ( 3-13) the resolution must at least 300 dpi

10-                     The authors must added response surface methodology to optimize

11-                     The authors must add mechanism for work

12-                     The authors must modify 3.10. conclusion to 4. conclusion

13-                     References must be updated by add page numbers and volume full data

Recommendation: Major revision

Comments on the Quality of English Language

minor revisions required

Author Response

Dear Reviewer

     Thank you immensely for your thorough review and invaluable feedback on our revised manuscript. Your insightful comments have greatly contributed to the enhancement of our work, and we sincerely appreciate the time and effort you have dedicated to evaluating our research.We have addressed each of your suggestions and concerns as following:

1-Title must be modified to remove abbreviation

A: We have already made the necessary changes to the title, removing any abbreviations as suggested.

2- In abstract, the authors should add problem at first of abstract

A: We have added relevant content at the beginning of the abstract, as well as highlighted it in blue, in accordance with your recommendation.

3- In abstract, authors should add more quantitative data

A: Additional quantitative data has been incorporated into the abstract, addressing your concern.

4- In abstract, the authors should add more data at least 250-300 words

A: We have expanded the abstract to meet the required word count, and the relevant content has been highlighted in Blue. Additionally, we have supplemented the introduction with the same content, located at page 3, lines 107-126.

5- In abstract, the authors must add prospective work at end of abstract

A: It has been added at end of abstract and we also have moved the prospective work to the conclusion section, at page 16,lines 546-553, as suggested. The content has been adjusted accordingly.

6- In abstract, the authors must add novelty in abstract

A: Further descriptions of novelty have been included in the abstract, specifically on page 2, lines 87-106.

7- Keywords must at least six words

A: We have added the required number of keywords and marked them accordingly.

8- Page 2 line 62,authors must move this into experimental

A: The relevant content has been adjusted and moved to section 2.4 (page 5) "Preparation of Composite Dressing Particles" under the experimental section. The description has been expanded accordingly. Additionally, the process and reaction mechanism have been relocated to section 3.1 "The PHMB Loading and Release Mechanism of the Dressing" under the results and discussion section at page7.

9- Figures ( 3-13) the resolution must at least 300 dpi

A:We have enhanced the resolution of all figures to meet the required minimum of 300 dpi.

10-The authors must added response surface methodology to optimize

A: We have added the response surface methodology for optimization in section 2.3 "Response Surface Methodology" at page 4 and the results analysis in section 3.2 "Analysis of Response Surface Optimization Results"at page 8

11- The authors must add mechanism for work

A:We have added section 3.1. "The PHMB Loading and Release Mechanism of the Dressing," including the synthesis mechanism and reaction equations, along with explanations.

12- The authors must modify 3.10. conclusion to 4. conclusion

A: Thank you for your suggestion. We have made the necessary adjustment.

13- References must be updated by add page numbers and volume full data

A: References have been updated newly with the addition of page numbers and full volume data.

      We appreciate your thorough review and valuable feedback on our manuscript. We have addressed all the issues raised and made the required modifications. If you have any further suggestions or concerns, we would be happy to consider them and make additional revisions as needed. We look forward to hearing from you.

Warm regards,

Heshuang Dai

Reviewer 2 Report

Comments and Suggestions for Authors

The manuscript aims to describe the development and evaluation of a novel antibacterial wound dressing  based on carboxymethyl cellulose ( CMC) and Pullulan Gel (Pul) and incorporating polyhexamethylene biguanide (PHMB) - a well-known antiseptic compound. Their proposed  dressing-materialswhich combined validate properties of composite dressing particles used, are investigated in terms of absorbency, rheological properties, PHMB loading and release behavior, and antibacterial activity. The healing effect was investigated comparing PHMB-loaded dressing with non-loaded dressing and untreated wounds.

Although the topic is interesting, there are several issues that have to be addressed:

1. Introduction

- All abbreviations must be decoded the first time they are mentioned (including in the abstract)

- The introduction has to be re-written / must be completed to include specific bibliography which supports the hypothesis the authors tested. The current format of the introduction is more suitable for a conclusion.

- There are not figures in the text of section 1- Introduction (There are no references to figures 1 and 2) in the text. 

- Two questions regarding Figure 2 (Preparation and application of two types wound dressing):

- Where is the illustration of the application?

- Fig.2 should be included in the introduction or in the experimental section 2?

2. Material and methods

- Is the in vivo study (1.1.4- In Vivo Experiments on Mice -) in accordance with the regulations regarding the animals treatment and did it have the necessary ethical approvals?

3. Results and discussion 

- The results of the study are predictable: it is known that CMC's chemical architecture lends it greater absorbency; the properties of hydrogels are also well known, explaining the Pul rheological properties; both materials proposedshowed significant antibacterial effects in antimicrobial assays (due to PHMB). What are the advantages compared to other antibacterial wound-dressings available in medical practice ?

In this study, the properties of both formulation have been assessed. All experiments compared Pul-PHMB/GB and CMC-295 PHMB/GB, with one exception: the study validates the clinical applicability only for Pul-PHMB/GP. The authors should discuss why.

Have the authors performed histological studies?Experiments should be included in this study

- The authors did not discuss their results in the context of the current literature. The authors should correlate their findings with already-published studies.

Minor revision: 

- Change Font size equation 1 and 2

-Change  Font in  Italic- Pseudomonas aeruginosa, Bacillus subtilisStaphylococcus aureus, as well as Escherichia coli (ex . line 195,  323-324, 330-331,  334 …) 

- Revise author instruction for correct references citation in the manuscript.

Author Response

Dear Reviewer

Thank you immensely for your thorough review and invaluable feedback on our revised manuscript. Your insightful comments have greatly contributed to the enhancement of our work, and we sincerely appreciate the time and effort you have dedicated to evaluating our research.

Below, we have addressed each of your suggestions and concerns:

1.Introduction

①All abbreviations must be decoded the first time they are mentioned

A: abbreviations have been  corrected, and can be confirmed in the original text.

②Rewriting of Introduction and Inclusion of Specific Bibliography

A: The introduction has been rewritten and improved to include specific literature supporting the hypotheses we tested. This includes descriptions of the mechanism and application of PHMB[5–10], as well as research literature on the characteristics of raw materials[18–21] and powder preparation[22]. By referencing these sources, we have provided a clearer exposition of the background and hypotheses of our study, positioning them in a more appropriate context. We hope this could meets your requirements.

③Confusion about Figure1 and Figure2

A: Figures have been supplemented with explanatory notes and corrections. Regarding Figure 1, references and brief explanations have been added, as seen in introduction page 2, lines 49. Figure 2 has been  moved to section 2.4 "Preparation of Composite Dressing Particles" in the Experimental section, with corresponding descriptions added at  Page 5.

 2.Material and Methods

①The confuse about the ethical approvals

A: The in vivo study had been conducted in accordance with animal treatment regulations and had received the necessary ethical approvals,detailed information as indicated in section 2.17 "Ethical Considerations in Animal Experimentation."at Page 7.

3.Results and discussion 

①Advantages Compared to Other Wound Dressings

A: Traditional dressings typically utilize gel formulations of CMC and Pul, which provide a moist environment for wound healing by aiding in cell migration. They could uniformly and continuously release PHMB, providing prolonged antimicrobial protection. However, for wounds with a large amount of exudate, gels may not be sufficient to absorb all the fluid, necessitating the use of additional absorbent materials.  Additionally, gel dressings may completely cover the wound, making wound observation difficult.

The main feature of our study is the preparation of dressings in the form of freeze-dried powder.   This form overcomes some limitations of liquid and gel dressings, particularly in storage and precise application. It prevents the premature expansion or dissolution of PHMB in high-humidity environments. This powder preparation could transform into a gel upon contact with wound exudate, providing lasting antimicrobial protection while reducing the frequency of dressing changes.  It ensures accurate coverage of irregular wounds, such as those from burns or pressure sores, and exhibits excellent chemical and physical stability in a dry state, facilitating storage and transportation.

we have incorporated this section into the new introduction to better elucidate the importance and innovation of our study.

②the reason for the study validates the clinical applicability only for Pul-PHMB/GP.

A: During our investigation, it was noted that the CMC-PHMB/GP  displayed pronounced hygroscopic properties. This observation aligns with the findings illustrated in Figure 6(page10), where CMC-gel exhibited a higher degree of swelling compared to Pul-gel. This property resulted in notable water absorption during storage, leading to the development of a sticky and unmanageable texture. Consequently, CMC-PHMB/GP proved unsuitable for formulation into a stable, usable medicated powder dressing. This finding is consistent with the excessively high Hausner ratio values of 2.71 observed for CMC-PHMB/GP in Table 2( page 11).

In contrast, Pul-PHMB/GP maintained a more controlled hydration level and mechanical stability, rendering it suitable for further development and clinical testing.

This discrepancy underscores the critical importance of selecting appropriate materials for powder preparation dressings, as they directly impact the clinical feasibility and effectiveness of the product.

The corresponding clarification has been brieflyprovided in section 3.11, "Evaluation in Model Mice," of the original manuscript.

③The Requirement of Histological Studies

A: The histopathological examination details are provided in section 2.16, "Histopathology" at page 7 of the experimental chapter.The results and discussion pertaining to this aspect have been included in section 3.11, "Evaluation in Model Mice" at page 16 towards the end of the page signed at blue color. Specific details of the results and analysis can be found in the newly added content in the revised manuscript.

Discussion in the Context of Current Literature

A: For this issue, I made the following revisions:

a. The introduction section mentions previous studies on the mechanism and application of PHMB [5–7], describing its antibacterial mechanism and its importance in wound management.

b. When introducing various PHMB-related dressings, the introduction section cites some published research literature as support. For example, references to the studies by O-chongpian et al. [15] and Laleh et al. [16] are included, where they respectively developed different types of PHMB dressings and highlighted their characteristics and limitations.

c. The introduction section also describes the characteristics and advantages of the novel powder-prepared wound dressing introduced in this study (Pul-PHMB/GP), and mentions its potential clinical applications.

I hope these could meets your requirements.

4. Minor Revisions

① Font Size and Style Corrections

A: Font size and style corrections have been made to equations and italicized text references.

② Revised Author Instructions for References

A: The entire manuscript has been revised for correct referencing.

We appreciate your thorough review and valuable feedback on our manuscript. We have addressed all the issues raised and made the required modifications. If you have any further suggestions or concerns, we would be happy to consider them and make additional revisions as needed. We look forward to hearing from you.

Warm regards,

Heshuang Dai

Round 2

Reviewer 1 Report

Comments and Suggestions for Authors

all comments are done

Reviewer 2 Report

Comments and Suggestions for Authors

The authors responded to my observations and the present manuscript has been improved. The revised manuscript is suitable for publication in its present form.

However, a few minor remarks (writing errors):

Line 171: 2.4. Pre2.4.Prepara2.4 Preparation of Composite

Line  497:    4. Conclusion (capitalized!)